# A Monolithic 3-Dimensional Static Random Access Memory Containing a Feedback Field Effect Transistor

**DOI:** 10.3390/mi13101625

**Published:** 2022-09-28

**Authors:** Jong Hyeok Oh, Yun Seop Yu

**Affiliations:** ICT & Robotics Engineering, Semiconductor Convergence Engineering, AISPC Laboratory, IITC, Hankyong National University, 327 Jungang-ro, Anseong-si 17579, Gyenggi-do, Korea

**Keywords:** monolithic 3-dimensional integrated, static random access memory, feedback field effect transistor, electrical coupling

## Abstract

A monolithic three-dimensional integrated static random access memory containing a feedback field effect transistor (M3D-FBFET-SRAM) was proposed. The M3D-FBFET-SRAM cell consists of one metal oxide semiconductor field effect transistor (MOSFET) and one FBFET, and each transistor is located on the top tier and one on the bottom tier in a monolithic 3D integration, respectively. The electrical characteristics and operation of the NFBFET in the M3D-FBFET-SRAM cell were investigated using a TCAD simulator. For SRAM operation, the optimum doping profile of the NFBFET was used for non-turn-off characteristics. For the M3D-FBFET-SRAM cell, the operation of the SRAM and electrical coupling occurring between the top and bottom tier transistor were investigated. As the thickness of interlayer dielectric decreases, the reading ‘ON’ current decreases. To prevent performance degradation, two ways to compensate for current level were suggested.

## 1. Introduction

Over the past few decades, computing systems have followed the von Neumann architecture [1]. An important feature in this architecture is the data process. The processed data are transferred from a processing unit to a memory unit. Throughout this process, intermediate memory storage is necessary. In the current computing systems, the volatile type of memory circuits that maintain data when the supply voltage is applied are configured close to the processing unit. Static random access memory (SRAM) is designed for the nearest processing unit because it performs very rapidly and does not need to be refreshed [2]. The conventional SRAM cell has two access transistors and two inverters, which consist of two transistors. Due to the SRAM circuit configuration, the cache memory made up of SRAM occupies a large portion of the overall processor chip. For designing next-generation processor chips, it is necessary to improve the performance and reduce the area of the SRAM.

A monolithic three-dimensional (M3D) integration is one of fabrication technologies for overcoming Moore’s Law [3,4,5,6,7,8,9]. The circuits with M3D structure are designed vertically for the transistor, logic gates, and system level. The circuit designed with M3D technology exhibits a high integration of the transistors and has a low propagation delay due to vertical interconnection. In order to utilize these characteristics, various circuits with the M3D structure were studied [10,11,12,13,14,15,16]. Particularly, the 6T (or more than six transistors) SRAMs with M3D structure have been proposed and studied to increase density, which can achieve up to a 45% density increase [17,18,19,20,21,22].

To improve the performance, density, and power consumption of SRAM, the configurations of SRAM with novel devices are introduced [23,24,25]. Among these, a feedback field effect transistor (FBFET) has been attracting attention for use in a next-generation memory device due to its steep slope and hysteresis characteristics [26]. Moreover, various memory circuits containing the FBFET have been proposed because the FBFET can be fabricated with a complementary MOS (CMOS) process. The SRAM consisting of one FBFET and one MOSFET (FBFET-SRAM) was proposed [27]. The density and power dissipation of FBFET-SRAM cell are improved more than those of the conventional 6T-SRAM. The FBFET-SRAM can be stacked vertically with the M3D structure because the devices used in the FBFET-SRAM, such as fully depleted silicon on insulator (FDSOI) FET, FBFET, junctionless FET, tunnel FET, gate-all-around FET, and nanosheet FET, are stackable. Therefore, there is a need for research on FBFET-SRAM with M3D structure (M3D-FBFET-SRAM) cells to meet the high density demands.

In this study, the M3D-FBFET-SRAM cell is proposed, and its electrical characteristics are investigated using technology computer aided design (TCAD). First, the simulation structure of the M3D-FBFET-SRAM cell is described in Section 2. In Section 3, the electrical characteristics of the M3D-FBFET-SRAM cell regarding DC characteristics and cell operation are discussed. In Section 4, the electrical coupling occurring at the top tier transistor is discussed. Finally, the conclusions of this study are described.

## 2. Simulation Structure

Figure 1a–c show the three-dimensional bird’s eyes view of the M3D-FBFET-SRAM cell, the process sequence, and a cross-section of the A-A` and circuit diagram of the M3D-FBFET-SRAM cell, respectively. The M3D-FBFET-SRAM cell consists of N-type FBFET (NFBFET) and N-type MOSFET (NMOSFET), and the NFBFET and NMOSFET are located on bottom tier and top tier, respectively. The material compositions and doping profile for the M3D-FBFET-SRAM cell were described, as shown in Figure 1a. The work-function of gate metal was used for 5.0 eV at NFBFET and NMOSFET. The device widths are 160 nm and 80 nm for the NFBFET and NMOSFET, respectively. The M3D-FBFET-SRAM cell was fabricated using a Victory Process simulator [28], and the electrical characteristics of the M3D-FBFET-SRAM cell were investigated using the commercial TCAD simulation program Atlas in mixed-mode [29]. The fabrication process was based on FDSOI technology [30] for the bottom and top tier transistors. In particular, the fabrication of the top tier transistor requires a low temperature in the monolithic fabrication technology due to interlayer dielectric (ILD) [31]. Therefore, the low temperature was used during the deposition, ion implantation, and annealing process, as shown in Figure 1b. Table 1 shows the structure parameters of the M3D-FBFET-SRAM cell. The physical models including CVT, SRH, and FERMI for NMOSFET, as well as CONMOB, FLDMOB, CONSRH, AUGER, and BGN for NFBFET, were used for simulation.

## 3. Simulation Results

In this section, the electrical characteristics of the M3D-FBFET-SRAM cell will be discussed. First, the operation and electrical characteristics of the NFBFET were investigated with respect to the energy band diagram. Moreover, during the writing operation of the M3D-FBFET-SRAM cell, the role of NFBFET was investigated. Based on the NFBFET operation, the M3D-FBFET-SRAM cell operation was investigated, particularly, due to the M3D structure, the electrical coupling occurring between the top and bottom tier transistors [32]. This coupling effect causes the electrical characteristics of the top tier transistor to change. Therefore, the investigation of the electrical coupling was conducted.

### 3.1. Electrical Characteristics of the NFBFET in the M3D-FBFET-SRAM Cell

Figure 2a–c shows the energy band diagram of the NFBFET under three different bias conditions. The red and black lines denote the conduction band and valence band, respectively. For the initial state of the NFBFET, the bit line voltage (*V_BL_*) and word line 2 voltage (*V_WL2_*) are applied with 1.9 V and 1 V, respectively. At this state, the electrons from the source region cannot be injected into the channel region, due to the potential barrier at the gated channel region, as shown in Figure 2a. When the forward sweep starts at the word line 1 voltage (*V_WL1_*), the potential barrier is lowering, and the electrons drift into the ungated channel region by the drain-source field. The injected electrons accumulate at the potential well at the ungated channel region; thereafter, the potential well is eliminated by accumulated electrons. The holes from the drain region can diffuse by the lowered potential barrier at the ungated channel region, and accumulate at the gated channel region, as shown in Figure 2b. Finally, this positive feedback between the electrons and the holes injection causes the energy band of all the regions align, as shown in Figure 2c. Figure 3 shows the drain-source current of the NFBFET (*I_DS-NFBFET_*) versus *V_WL1_*. There is an abrupt increment of the NFBFET current at *V_WL1_* = 0.17 V. The hysteresis characteristic, which is the threshold voltage difference between forward and reverse, can be controlled by the doping profile of the channel region [33]. For the FBFET-SRAM operation, the very large memory window or non-turn-off characteristics by the gate-field, are required for maintaining the reading ‘ON’ current level, as shown in Figure 3. The doping profile is adjusted to satisfy the performance of NFBFET.

Figure 4a,b shows the NFBFET current (*I_DS-NFBFET_*) for writing the ‘ON’ and ‘OFF’ operation in the M3D-FBFEET-SRAM cell, respectively. The red line and black square-lines denote the operation current and DC characteristics of the NFBFET, respectively. When the writing ‘ON’ operation begins, *V_BL_*, *V_WL1_*, and *V_WL2_* are applied for 1.9 V, 0.6 V, and 1.0 V, respectively. When the writing ‘ON’ pulse is applied to the M3D-FBFET-SRAM cell, the NFBFET currents change following the blue arrows, as shown in Figure 4a. When the writing ‘OFF’ operation begins, *V_BL_*, *V_WL1_*, and *V_WL2_* are applied for 0.7 V, 0.6 V, and 1.0 V, respectively. When the writing ‘OFF’ pulse is applied to the SRAM cell, the NFBFET current changes following the blue arrows, as shown in Figure 4b.

### 3.2. M3D-FBFET-SRAM Cell Operation

Figure 5 shows the timing diagram of the M3D-FBFET-SRAM cell operation. The black, blue, green, and red lines denote the voltage pulse of *V_BL_*, *V_WL1_*, and *V_WL2_*, and the current pulse of *I_DS-NFBFET_*, respectively. Table 2 shows the M3D-FBFET-SRAM cell operation voltages. The rising, falling, and pulse-width times are 0.2 ns [27]. For the reading ‘ON’ and ‘OFF’ currents, *I_DS-NFBFET_* are approximately 15 μA and 0.2 nA, respectively. The writing ‘ON’ and ‘OFF’ speeds are about 0.4 ns, as shown in Figure 5. For the first suggested FBFET-SRAM, the unit cell size is 8F^2^ (F = feature size) [27], and this cell achieves very high density compared with conventional 6T-SRAM. However, when the FBFET-SRAM is designed with the M3D structure, the cell area can decrease up to 50% compared with the planer 2-D cell structure [27].

Figure 6a,b show the retention characteristics of the M3D-FBFET-SRAM cell when the recursive reading pulse is applied after the writing ‘ON’ and ‘OFF’ operations, respectively. For investigating the retention characteristics of the M3D-FBFET-SRAM cell, the retention time was 30,000 s. When the NFBFET is turned on, the NFBFET remains in the on-state until it has formed the potential well in the channel region. For maintaining the on-state, the NFBFET requires appropriate *I_DS_-_NFBFET_*, which can be controlled by holding voltages of *V_WL2_* [27]. As shown in Figure 6a,b, the M3D-FBFET-SRAM cell maintains the data for 30,000 s after the writing ‘ON’ and ‘OFF’ operations, respectively.

### 3.3. Electrical Coupling

For the M3D structure, the electrical characteristics of the top tier transistor can be changed by the thickness of the interlayer dielectric (*T_ILD_*). As *T_ILD_* decreases, the electric field occurring at the bottom tier transistor effects the top tier transistor. This coupling effect causes unexpected changes in system performance. In order to design the system with intended performance, the investigation for changing the electrical characteristics must be proceeded with respect to *T_ILD_*. In this section, the electrical coupling occurring between the top and bottom transistors was investigated. Furthermore, the optimum voltages were suggested for short *T_ILD_*.

Figure 7a shows the drain-source current of the NMOSFET (*I_DS-NMOSFET_*), which is located in the top tier, at *T_ILD_* = 3 nm. The red and black lines denote *I_DS-NMOSFET_* at *V_WL1_* = 0.6 and 0 V, respectively. As *T_ILD_* decreases, the electric field applied to the top tier transistor is stronger. When *V_WL1_* is applied for 0.6 V, the threshold voltages of *I_DS-NMOSFET_* changes from 0.67 V (for *V_WL1_* = 0 V) to 0.50 V, as shown in Figure 7a. Figure 7b shows the M3D-FBFET-SRAM cell operation at various *T_ILD_*. As *T_ILD_* decreases, the threshold voltage of *I_DS-NMOSFET_* decreases and *I_DS-NMOSFET_* increases at the same *V_WL2_*. The current level (*I_DS-NFBFET_*) must be lower to match the current levels of the NFBFET and the NMOSFET by their serial connection; thus, *V_A_* must also be reduced, as shown in Figure 1c. Therefore, as *T_ILD_* decreases, the reading ‘ON’ currents decrease in the direction of the orange arrow, as shown in Figure 7b.

Figure 8a,b shows the reading ‘ON’ current of the M3D-FBFET-SRAM cell with *T_ILD_* = 3 nm at modified *V_BL_* and *V_WL2_*, respectively. The black and red lines denote *I_DS-NFBFET_* with original and modified voltage levels, respectively. To design the M3D-FBFET-SRAM with shorter *T_ILD_*, it is necessary to solve the problem of lowering the reading ‘ON’ current, because the low reading ‘ON’ current is a critical problem in SRAM operation. To solve this problem, higher *V_BL_* and *V_WL2_* can be chosen to create a high reading ‘ON’ current, as shown in Figure 8a,b. The shorter T_ILD_ can achieve a lower critical delay due to shorter monolithic inter-tier via the *T_ILD_*. However, higher voltage levels of *V_BL_* and *V_WL2_* are required, and power consumption will be high. In order to design the M3D-FBFET-SRAM, the appropriate *T_ILD_* must be investigated for an acceptable trade-off regarding the performance.

## 4. Conclusions

In this study, the M3D-FBFET-SRAM cell was proposed. The M3D-FBFET-SRAM cell consists of one NFBFET and one NMOSFET, and the NFBFET and NMOSFET in the M3D structure are located on the bottom and top tier, respectively. The transistors are stacked vertically; therefore, the M3D-FBFET-SRAM cell area achieved up to a 50% reduction compared with 2-D planer cell structure. The electrical characteristics of the M3D-FBFET-SRAM were investigated. First, for the NFBFET, the DC characteristics and role were investigated during the writing operation. In order to achieve the non-turn-off characteristics, the optimum doping profile of the NFBFET was used. Based on the NFBFET operation, the M3D-FBFET-SRAM cell operation was investigated. The reading ‘ON’ and ‘OFF’ current are approximately 15 μA and 0.2 nA, respectively. For the retention characteristics, the M3D-FBFET-SRAM cell can maintain the data for at least 30,000 s. In particular, the electrical coupling occurring between the top and bottom transistors was investigated with respect to *T_ILD_*. As *T_ILD_* decreases, the reading ‘ON’ current decreases. To compensate for the reading ‘ON’ current, a higher *V_BL_* and *V_WL2_* can be applied; however, the power consumption will be higher. Therefore, in order to design the M3D-FBFET-SRAM, the investigation of the appropriate *T_ILD_* must be conducted.

## Figures and Tables

**Figure 1 micromachines-13-01625-f001:**
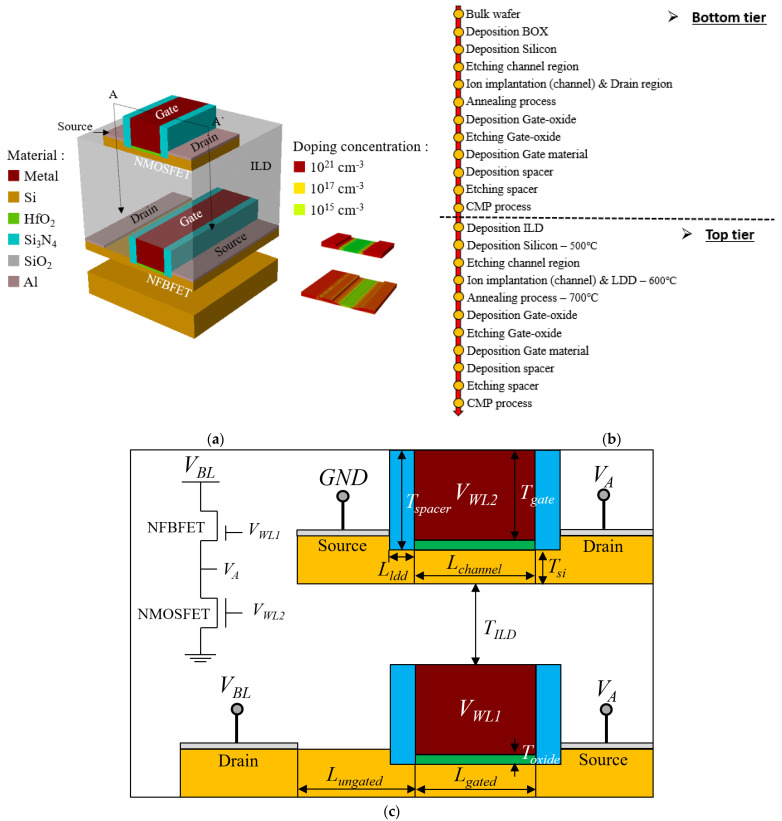
(**a**) A three−dimensional bird’s eyes view of the M3D−FBFET−SRAM cell, (**b**) its fabrication process sequence, (**c**) its cross−section of A-A` and circuit diagram.

**Figure 2 micromachines-13-01625-f002:**
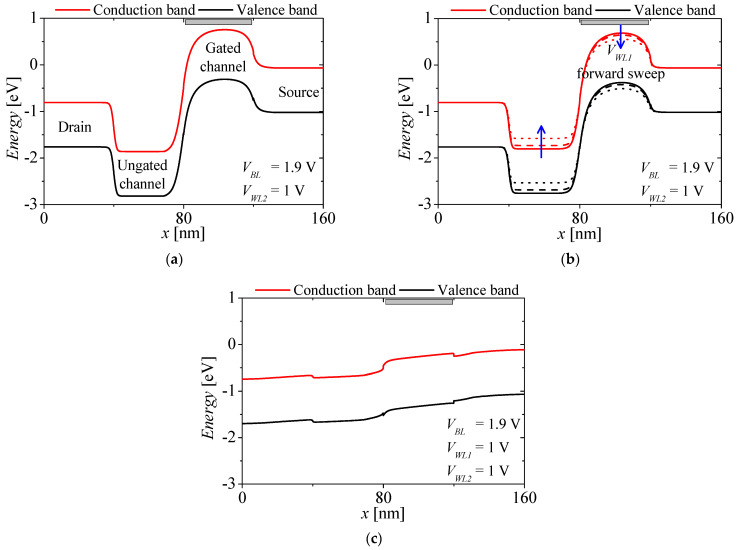
The energy band diagram of the NFBFET in the M3D−FBFET−SRAM cell. (**a**) Initial state (*V_BL_* = 1.9 V, *V_WL2_* = 1 V, and *V_WL1_* = 0 V), (**b**) *V_WL1_* forward sweep (*V_BL_* = 1.9 V and *V_WL2_* = 1 V), (**c**) on−state (*V_BL_* = 1.9 V and *V_WL1_* = *V_WL2_* = 1 V).

**Figure 3 micromachines-13-01625-f003:**
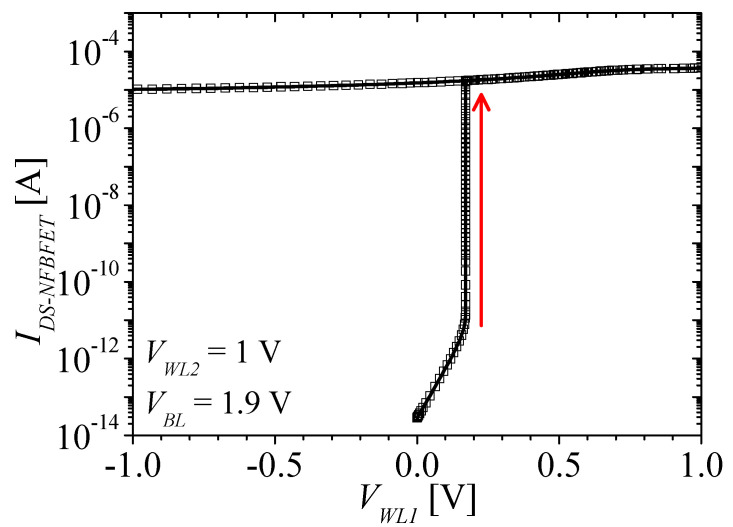
Drain−source current of the NFBFET (*I_DS−NFBFET_*) versus *V_WL1_* when *V_BL_* = 1.9 V and *V_WL2_* = 1 V.

**Figure 4 micromachines-13-01625-f004:**
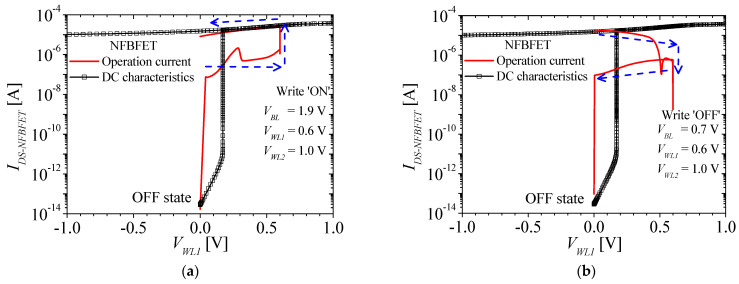
The NFBFET current during the (**a**) writing ‘ON’ operation and (**b**) writing ‘OFF’ operation.

**Figure 5 micromachines-13-01625-f005:**
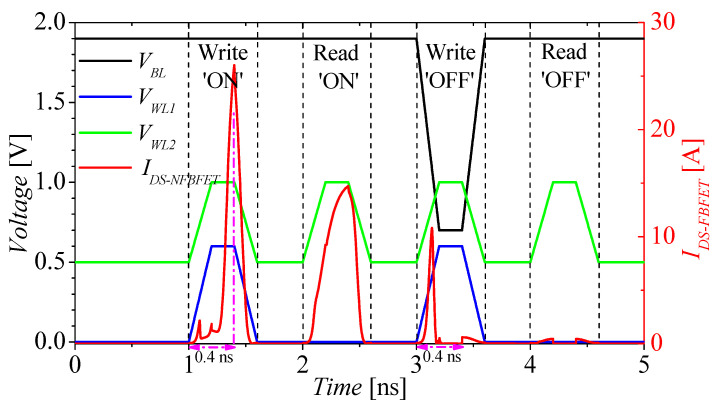
Timing diagram of the M3D−FBFET−SRAM cell operation.

**Figure 6 micromachines-13-01625-f006:**
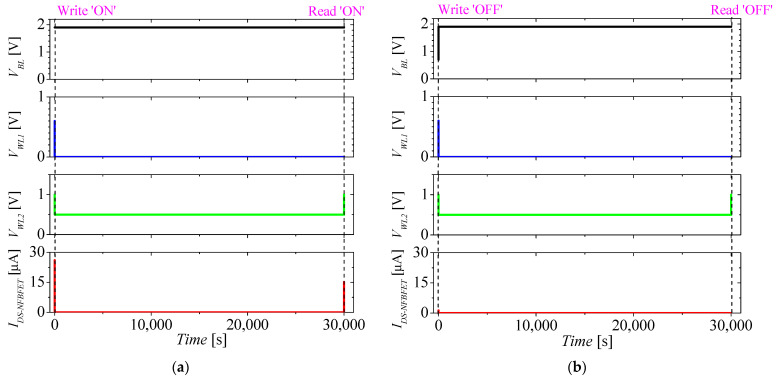
Retention characteristics of the M3D−FBFET−SRAM cell when the recursive reading pulse is applied after (**a**) writing ‘ON’ operation, and (**b**) writing ‘OFF’ operation.

**Figure 7 micromachines-13-01625-f007:**
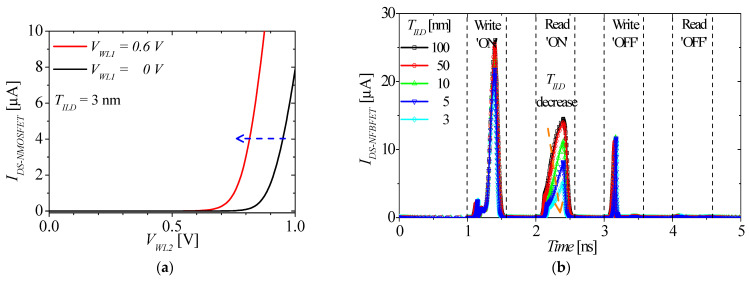
(**a**) Changing threshold voltage at two different bottom transistor gate voltages (*V_WL1_* = 0 and 0.6 V) at *T_ILD_* = 3 nm. (**b**) The M3D−FBFET−SRAM cell operation with respect to various *T_ILD_*.

**Figure 8 micromachines-13-01625-f008:**
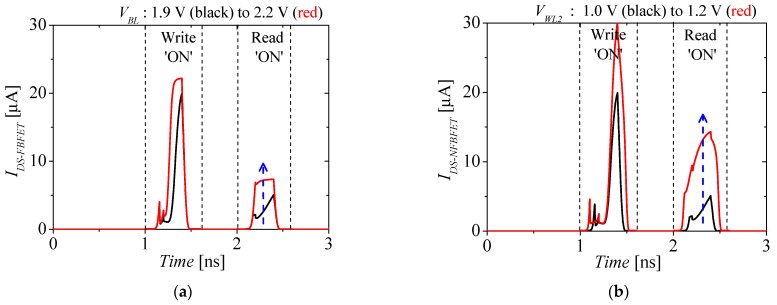
The reading ‘ON’ current of the M3D−FBFET−SRAM cell with *T_ILD_* = 3 nm. (**a**) *V_BL_* = 1.9 (black lines), and 2.2 V (red lines), and (**b**) *V_WL2_* = 1.0 (black lines) to 1.2 V (red lines).

**Table 1 micromachines-13-01625-t001:** Structure parameters of the M3D-FBFET-SRAM cell.

Parameters	Description	Value/Unit
*L_ungated_*	Length of the ungated channel region for the NFBFET	40 nm
*L_gated_*,* L_channel_*	Length of the gated channel region for the NFBFET and NMOSFET	40 nm
*L_LDD_*	Length of the lightly doped drain (LDD) region	10 nm
*T_gate_*	Thickness of the gate	30 nm
*T_spacer_*	Thickness of the spacer	33 nm
*T_oxide_*	Thickness of the gate oxide	3 nm
*T_si_*	Thickness of the silicon body	6 nm
*T_ILD_*	Thickness of the interlayer dielectric (ILD)	Var.
*N_source_*,* N_drain_*	Doping concentration of the source and drain regions	1 × 10^20^ cm^−3^
*N_gated_*	Doping concentration of the gated channel region for NFBFET	2 × 10^17^ cm^−3^
*N_ungated_*	Doping concentration of the ungated channel region for NFBFET	1 × 10^20^ cm^−3^
*N_ch_*	Doping concentration of the channel region for NMOSFET	1 × 10^15^ cm^−3^
*N_LDD_*	Doping concentration of the LDD region for NMOSFET	1 × 10^18^ cm^−3^

**Table 2 micromachines-13-01625-t002:** The M3D−FBFET−SRAM cell operation voltages.

Voltages	Write ‘ON’	Write ‘OFF’	Hold	Read
*V_BL_*	1.9 V	0.7 V	1.9 V	1.9 V
*V_WL1_*	0.6 V	0.6 V	0.0 V	0.0 V
*V_WL2_*	1.0 V	1.0 V	0.5 V	1.0 V

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
