# Peer review of "A Monolithic 3-Dimensional Static Random Access Memory Containing a Feedback Field Effect Transistor"

_micromachines, 2022, doi:10.3390/mi13101625_

Round 1
Reviewer 1 Report
This paper proposes a 2T SRAM via a FBFET and a NMOS, stacked in a 3D fashion. This paper is overall written well and interesting, but I have a few comments :
1. SRAM should be able to hold data indefinitely as long as the supply is available. I don’t see how this cell can hold its value indefinitely; eventually, the charge in the NFBFET will still leak out. As a result, I consider it a DRAM with a long retention time. If the proposed cell is a SRAM, please explain how it can hold charge indefinitely.
2. Has the leakage from the NFET been considered? The NFET should leak out the charge stored in the NFBFET fairly quickly. Even in DRAM with a large cap, optimized transistor leakage, and negative gate voltage bias the charge in DRAM is gone after 5ms. The Charge stored in the NFBFET is much smaller than DRAM, the NFET transistor is not optimized for leakage, and the gate is biased at 0. I don’t see how the proposed cell can hold charge longer than DRAM. Please explain this.
Author Response
Please find an attached file.
Yun Seop Yu

Reviewer 2 Report
The authors have presented work on FBFET based monolithic SRAM. The work is interesting and timely. I would recommend the work given that the authors clarify a few concerns. It is better if the authors add a proper circuit showing the write line and bit line for better clarity for the reader. Kindly calculate the signal-to-noise margin for read and write operation for better comparison with the state-of-the-art memory devices. Kindly clarify if the simulations are done on atlas or victory tools of silvaco.
Author Response

(The authors gave the same response as above.)

Round 2
Reviewer 1 Report
I have no further comments
Reviewer 2 Report
I would recommend the work be published as the authors have clarified most of the concerns.